# Identity fusion can foster intergroup trust and willingness to cooperate
Jack W. Klein [1,2] ✉, Brock Bastian [2], Emmanuel N. Odjidja[3], Samhita S. Ayaluri[3], Christopher M. Kavanagh [4], Alimudin M. Mala[5] & Harvey Whitehouse [4]

Identity fusion – a construct that captures extreme ingroup commitment – has traditionally been associated with intergroup violence. However, recent research suggests that identity fusion is also associated with feelings of security that promote intergroup interactions. This apparent contradiction was explored by examining moderators of the relationship between identity fusion and positive intergroup relations across two studies. Study 1, a pre-registered study on intergroup relations in the turbulent Bangsamoro region of the Philippines ($N = 816$), found that identity fusion was positively associated with outgroup trust when the outgroup was perceived positively. Study 2 ($N = 1576$) replicated these results across Gambia ($n = 236$), Pakistan ($n = 505$), Tanzania ($n = 337$), and Uganda ($n = 498$), while also finding that perceptions of the relationship itself (e.g., whether cooperation was judged beneficial to the ingroup) similarly moderated the effect of identity fusion on willingness to cooperate. These results suggest that identity fusion can have positive consequences for intergroup relations, depending on contextual perceptions.

## Identity fusion can foster intergroup trust and willingness to cooperate

Intractable intergroup conflicts are a major cause of violence, death, and destruction across the world. Age-old conflicts, such as the Arab-Israeli and Turkish-Kurdish conflicts, have persisted into the 21st century and are assumed to be exacerbated by extreme ingroup commitments. While at some level this is undoubtedly true – after all, it would be difficult to imagine an Arab-Israeli conflict in which its participants were not strongly aligned with either Arabs or Israelis – it is questionable whether ingroup commitment invariably increases the likelihood of intergroup conflict. The present article examines whether identity fusion, an extreme form of group alignment, might also be associated with positive intergroup relations under certain circumstances. Specifically, we seek to elucidate the circumstances in which identity fusion, as intriguing new theoretical developments suggest[1], promotes intergroup trust and willingness to cooperate.

## Identity fusion and intergroup relations

The concept of identity fusion has emerged in recent years as a prominent predictor of intergroup conflict. Identity fusion is a powerful form of ingroup commitment that involves a visceral sense of oneness with the group[2,3] and is strongly associated with a willingness to fight and die for the group[4,5]. It is distinguished from other measures, such as *identification* from social identity theory, via four principles: the *relational ties principle* (the intensely strong bond between fused group members), the *identity synergy principle* (the synergistic union between the self and the group), the *irrevocability principle* (the alignment with the group is long-lasting and stable), and the *agentic-personal-self principle* (fused people feel empowered and in control of the group)[6]. Practically, however, it is often distinguished via its propensity to strongly predict extreme and violent pro-group behavior. Fusion has been implicated in numerous intergroup conflicts, from farmer-herder fighting in Cameroon[7] to religious and ethnic terrorism[8]. The theory proposes that *fused actors*, whether through shared experiences, norms, or genetics[5], come to regard one another as family members, which then compels them to violently defend the group and its members when necessary[9]. For instance, almost half of all frontline fighters in the 2011 Libyan revolution reported feeling more fused to their battalion than their own families[10]; indeed, it is the perception of familial bonds between fused actors that drives them to fight for the ingroup and one another[11]. Nevertheless, studies have demonstrated that the relationship between fusion and outgroup hostility is moderated by perceptions of the outgroup, such as whether they are considered a threat[12–14] or dehumanized[15], with fusion predicting violence specifically against threatening outgroups and not others[16]. While fusion can certainly drive intergroup violence, perceptions of the outgroup appear to at least partially dictate whether this occurs.

[1]Hong Kong Institute of Asia-Pacific Studies, Chinese University of Hong Kong, Sha Tin, Hong Kong. [2]Melbourne School of Psychological Sciences, University of Melbourne, Melbourne, Australia. [3]Global Community Engagement and Resilience Fund, Geneva, Switzerland. [4]Institute of Cognitive and Evolutionary Anthropology, University of Oxford, Oxford, UK. [5]Consortium of Bangsamoro Civil Society, Cotabato City, Philippines. ✉e-mail: jack.klein@cuhk.edu.hk

Recent theorizing suggests that when perceptions of outgroups are more positive, fusion may actually increase a willingness to trust and cooperate with outgroups. The *fusion-secure base hypothesis* argues that a tightly bonded fused group – in which group members feel cared for, listened to, and protected – acts as a secure base from which intergroup exploration can occur[1]. In fact, the four defining principles of identity fusion ultimately describe a group alignment that is stable, agentic, comforting, and empowering; while originally used to explain the group dynamics that motivate extreme violent pro-group behavior, in more positive contexts these same principles could inspire intergroup trust and cooperation. This differs from similar intergroup work based on the social identity perspective[1][7], which has argued that increased adherence to shared norms and group-oriented behavior drive exploratory motivations in highly-identified people, while building on related work from the *social cure* literature[18] and attachment theory[19] that similarly emphasizes the propensity for intimate groups to motivate intergroup contact. Empowered by this base, fused actors are free to trust and cooperate with outgroup members for the benefit of the self and the ingroup; however, should the outgroup come to be perceived negatively or as a threat, fused actors will be motivated to violently defend the ingroup[1]. Empirical evidence suggests that fusion is associated with a willingness to trust and interact with others, including outgroup members[20][21], if considered unthreatening[22]. More generally, the fusion-secure base hypothesis suggests that a fused group empowers individuals to undertake whatever behaviors are judged most advantageous to the ingroup[1]. Cooperation and trust are hence most likely when mutual benefit is salient, while the avoidance of cooperation will be preferred if interactions with the outgroup are likely to negatively affect the ingroup.

The present study seeks to test the moderation component of the fusion-secure base hypothesis in naturalistic settings, in which outgroup perceptions vary, thereby illustrating the circumstances under which identity fusion elicits trust and willingness to cooperate. In doing so we illustrate that identity fusion can have positive outcomes in the right social context, counterbalancing the prevailing focus of the identity fusion literature on its contribution to intergroup conflict.

## Study Overview

Study One was a pre-registered test of the model in the Bangsamoro Autonomous Region of Muslim Mindanao (BARMM), a conflict-prone region of the Philippines with multiple conflicting ethnic, religious, and political intergroup dyads[23]. For example, the Moro (*13 predominantly Muslim ethnolinguistic groups originating in the BARMM*) have waged a low-intensity conflict against the Christian Settlers (*immigrants from elsewhere in the Philippines*) since the 1940s, while both groups have been in conflict with the Lumad (*indigenous inhabitants of the BARMM*) primarily over land[24]. These disputes often reflect religious divides, such as the destructive 5-month 2017 Battle of Marawi waged by local Islamic Jihadist groups[25]. Even political parties such as the Bangsamoro Party (BAPA) and United Bangsamoro Justice Party (UBJP) – which are aligned to rival militant groups – occasionally have violent clashes[26]. Differences in the contemporary state of these intergroup relations allow us to test the relationship between identity fusion and outgroup trust under varying intergroup contexts. First, we pre-registered the hypothesis that identity fusion predicts an openness to cooperate with outgroups in general. Reflecting the prominent role of threat in the fusion literature, we also pre-registered that the relationship between fusion and willingness to trust a specific outgroup would be moderated by outgroup threat, such that fusion would positively predict outgroup trust when perceptions of that outgroup as a threat were low. As an exploratory hypothesis, we also expected that general perceptions of the outgroup (i.e., whether it was perceived warmly or coldly) would similarly moderate this relationship.

Study Two was a conceptual replication and expansion of our model across four countries – Pakistan, Tanzania, Uganda, and Gambia – focusing on religious intergroup relations. This time we focused on a new dependent variable – willingness to cooperate with an outgroup – and a broader range of moderators. Intending to replicate Study 1, we predicted that fusion to

religion would predict a willingness to cooperate, but only when the outgroup was perceived positively. Similarly, as a more subtle measure of threat, we asked participants whether the ingroup has historically suffered from the behavior of the outgroup. As an extension, we explored whether perceptions of the likely outcomes of cooperation would moderate the relationship between fusion and willingness to cooperate. Specifically, we predicted that fusion would be a stronger predictor when cooperation was judged as beneficial to the ingroup, and a negative predictor when judged as likely to harm the ingroup (i.e., interactions with the outgroup were perceived as zero-sum). We also examined whether identity fusion and identification exhibited differing interactive patterns with the various moderators.

## Methods

### Study one

**Participants.** An a-priori sample size calculation was conducted using G-Power V.3.1 for a moderation effect with six predictors (power = 0.80; $\alpha$ = 0.05). Assuming a conservative small effect size ($f^2$ = 0.02), each interaction effect required 395 participants; there was hence a quota of at least 200 participants for each regional group. Trained enumerators surveyed Filipinos aged over 18 in pre-determined locations across the BARMM (see Fig. 1). To ensure a geographically diverse sample, surveys were conducted in 2–3 *barangays* (i.e., suburbs) per municipality, with a total of ten municipalities across the BARMM's two provinces, Maguindanao and Lanao del Sur. 817 participants were recruited overall, with 1 participant removed for reporting an age under 18 years. This final sample was 816 ($M_{age}$ = 36.67, $SD$ = 14.09), with an approximately even split of self-reported gender (390 men; 426 women).

**Ethics and inclusion statement.** Ethics approval for both studies was obtained from the University of Oxford Central University Research Ethics Committee, with ethical approvals obtained within each country as needed according to the standards and practices applicable. Specifically, local approval was obtained from the Uganda National Council for Science and Technology at Makerere University, Kampala and the Bio-Ethical Committee at Karakorum International University. For the Gambian fieldwork we followed Oxford's CUREC protocol under a collaboration agreement with North Carolina State University (where the Gambian expert was based).

Local researchers were either included as an author, or mentioned in the acknowledgements, depending on their overall contribution to the present research, with general roles and responsibilities agreed to ahead of the research. Local researchers helped with elements of study design, particularly in Study 1 in which advice was provided regarding which ingroups were most likely to be meaningful to participants. Study 1 had local relevance in that it helped gauge current intergroup attitudes, and was conducted as part of an ongoing capacity-building relationship between the local *Consortium of Bangsamoro Civil Society* and the *Global Community Engagement and Resilience Fund*, both of which are focused on reducing violence in the area. Study 2 was intended as a multi-country study to examine whether the results from Study 1 generalized and hence had less local relevance to any individual local context. The research was not expected to involve any distress or personal risk to participants, although enumerators in Study 1 were paired with participants of the same regional group (e.g., Moro enumerator with Moro participant) whenever possible to increase participant comfort. Also, local researchers for Study 1 completed a detailed risk assessment to mitigate any potential risks to enumerators. We have cited local research wherever relevant e.g.,[24].

**Materials and methods.** We initially conducted focus groups with BARMM residents to help write and refine the survey. The survey was translated from English to Tagalog by native Tagalog speakers, and then back translated to confirm the original meaning was retained. Data was collected in July and August 2022. Participants entered their own responses into tablets, with guidance from enumerators (typically from the same regional group) as required. This was to provide participants

**Fig. 1 | Enumerators administering the survey across the BARMM. a** An enumerator assists a participant complete the survey; **b** Enumerators complete a river-crossing to reach a remote *barangay*; **c** Enumerators enter a village to survey the inhabitants.

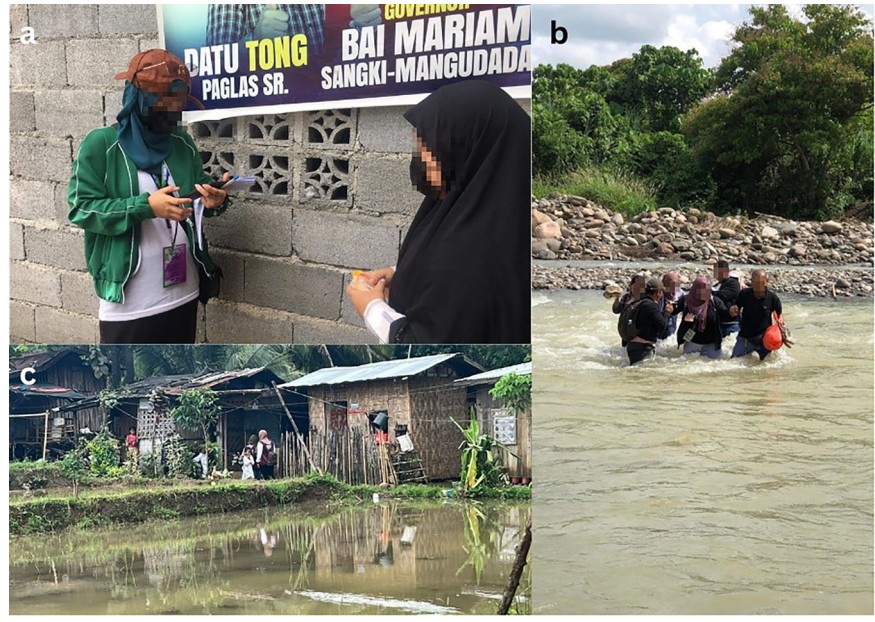

maximum privacy, given face-to-face interviews on sensitive issues have been associated with increased social desirability[27]. After providing informed consent, participants completed the scales outlined below.

This study was pre-registered on the 29th of August 2022 (see OSF for pre-registration: https://osf.io/h7svn/?view_only=960ebac75dba4e98 987a1693568e9554). We were unable to test H1 (i.e., identity fusion predicts a willingness to inflict harm on outgroups if they are assessed to be a threat, the ingroup accepts violence as a strategy, and peaceful strategies are not salient) as pre-registered, as our outgroup harm measure had insufficient variance. As an exploratory alternative we used social distance as the outcome measure (see Supplementary Note 1, and Tables S4 & S5). Analyses were also conducted using R (rather than SPSS as pre-registered) to ensure that our analyses were more easily reproducible, with all code supplied on the OSF.

First, participants reported which groups they most belonged to across multiple categories: religious (Muslim or Christian Filipinos, with specific denominational options for both), regional (Moro, Lumad, or Christian Settler), and political (UBJP or BAPA supporter). See Table S1 for the number of participants that belonged to each group. All ingroups were identified through consultation with residents of the BARMM and relevant experts, with tensions between these groups judged to be a key driver of intergroup violence in the region. These ingroup identities have considerable overlap; for example, almost all Moro are Muslim and Christian Settlers are Christian, with even UBJP and BAPA strongly tied to religious and ethnic identities. Some participants erroneously indicated that their group membership was *other* when in fact it fit into a provided category; these people were recoded into the appropriate category whenever it was clear that a mistake had occurred (e.g., participants who indicated they were not Christian but were instead Catholic were recoded as Christians). Participants then completed a pictorial measure of identity fusion[28] for each nominated ingroup. Participants chose which of several pictures of overlapping circles best represented their current relationship with an ingroup; if the circle representing the self was entirely encapsulated by that representing the group, the participant was judged to be fused. Measures of identity fusion (both the pictorial measure and the verbal measure used in Study 2) have been used in a range of cultural contexts, including in both interdependent and independent cultures e.g., [9].

Participants then reported the extent to which they trusted members from each group, and their general perception of the outgroup (i.e., warm vs. cold)[29]. Threat was also more explicitly measured by asking how much each participant felt an outgroup threatened "you and your loved ones" (i.e.,

realistic threat) and "you and your loved one's values" (i.e., symbolic threat)[30]. These items were averaged, with Cronbach's alphas ranging from 0.73 to 0.86. Openness to cooperation was measured with two items[31] asking whether participants felt their regional ingroup (e.g., Moro) could work with others not from their regional group to protect and benefit The Philippines. However, this measure was generic, with no explicit outgroups identified. All measures used a 7-point scale, excluding general outgroup perception which was measured on a 100-point scale (0 = Cold, 100 = Warm).

**Analytic strategy**. Analyses for both studies were conducted using *R Studio* version 4.3.1. We used a predefined level of significance of $p < 0.05$ for all statistical tests and report two-tailed significance levels.

A Welch's Two Sample t-test was used to compare fused and non-fused participants on openness to cooperation. All intergroup dyadic regression models focused on a single dyad (i.e., an ingroup and its corresponding outgroup) at the same level of abstraction (e.g., Christian Filipinos vs. Muslim Filipinos). All dyad ingroups were mutually exclusive (i.e., a participant could not belong to both ingroups). Each model included terms for fusion to the ingroup, dyad-specific group membership (e.g., if the participant was a Christian Filipino or Muslim Filipino), and the moderator main effect and interaction term. We included group membership to control for the baseline trusting tendencies of the ingroup, irrespective of fusion. Continuous independent variables involved in moderation analyses were scaled to facilitate interpretability. We calculated unstandardized regression coefficients for each model term, and adjusted $R^2$ for the overall model, to act as effect size estimates.

Assumption checking (including checking for normality and equal variances) indicated evidence of minor non-normality of residuals in some models, which was accounted for by repeating all analyses with bootstrapping. This had a negligible effect on all models but Model 5 (i.e., fusion to political party), in which the $p$-value for the moderation effect increased from 0.038 to 0.05. Given this lies on the bounds of statistical significance, the interaction from Model 5 should be interpreted with some caution.

**Study two**

**Participants.** Our initial sample was $N = 1594$, with a target of at least $n = 200$ each from four countries (Gambia, Pakistan, Tanzania, and Uganda). 18 participants were removed due to excessive non-response on the target items. A further 8 participants were identified as potentially low-quality due to straightlining; however, as sensitivity analyses suggested their inclusion did not affect results, they were retained. The final

sample was 1576 participants ($M_{age}$ = 36.57, $SD$ = 13.86) with a near equal split of self-reported gender (824 men; 715 women). This sample included the following number of participants from each country: Gambia ($n$ = 236), Pakistan ($n$ = 505), Tanzania ($n$ = 337), and Uganda ($n$ = 498). Our sample contained a range of ethnicities, with the most represented in each country's sample as follows: the Mandinka (Gambia), the Punjabis (Pakistan), the Maasai (Tanzania), and the Jopadhola and Iteso (Uganda).

**Materials & methods.** The survey was delivered in different languages, depending on the advice of local experts: English (all countries), Urdu (Pakistan), and Swahili (Tanzania). All translations were cross-referenced with the English version, and local enumerators provided explanations in local dialects as needed. Participants were paid at a rate equivalent to approximately £10 per hour in their local currency, except for Pakistan in which individual payments were not permitted and so all participants were volunteers. The survey was conducted using paper and pen. In the handful of cases a participant's intended answer could not be interpreted from their written response, it was disregarded. If participants wrote two consecutive numbers for a single answer these scores (e.g., 3 & 4) were averaged (e.g., 3.5), but if they wrote two non-consecutive numbers (e.g., 1 & 6) their answer was removed.

After providing informed consent, participants were asked if they considered a pre-determined religious group as a relevant ingroup. If not, they were asked to provide an alternative ingroup, although few participants did this. Participants also identified a religious outgroup using the same method. We focused on religious groups because previous research has suggested that they are commonly regarded as the most important ingroups (excluding family) in the target countries[32].

Next, participants completed scales related to their nominated ingroup and outgroup. Identity fusion was measured with reference to their ingroup using the verbal scale ($\alpha$ = 0.76), a more robust version of the fusion measure used in Study 1[33]. Although the scale consisted of four items, an error meant that one item was not collected in Uganda. To be consistent we excluded that item from the fusion scale across all countries. The remaining three items were: "I have a deep emotional bond with the [ingroup]", I am strong because of the [ingroup]", and "I make the [ingroup] strong". Participants also completed a one-item measure of identification "I identify with my group"[34], as well as the following three items: (1) "I have a lot in common with the ingroup", (2) "I connect with the values of the ingroup", and (3) "I feel a sense of belonging with the ingroup". When the identity fusion and these items were entered in an exploratory factor analysis all items loaded onto a "fusion" or "identification" factor, excluding the "I feel a sense of belonging with the ingroup". This item was dropped, while the others formed an identification scale ($\alpha$ = 0.75).

Participants then answered several questions related to perceptions of a specific outgroup. An explicit threat measure comprised of a single item asking participants whether they agreed that "The [outgroup] wants to harm the [ingroup]", while perceptions of the outgroup as a historical threat was measured with the item "Historically, the [ingroup] has suffered from the behavior of the [outgroup]". We also used the "General Evaluation Scale" measure ($\alpha$ = 0.91) which measures general outgroup perceptions[35]. This 5-item scale asked participants to place their feelings about the outgroup on a continuum between adjective pairs, ranging from a negative to a positive evaluation (e.g., "hostile-friendly", "threatened-relaxed", "concerned-unconcerned"). Perceptions that cooperation could benefit the ingroup was measured with the item "The [ingroup] can benefit from working with the [outgroup]", and perceiving the intergroup relationship as zero-sum was measured with "The [outgroup]'s gains are the [ingroup]'s losses". Finally, participants reported their willingness to cooperate with the outgroup with the item "I am willing to work with the [outgroup]". All measures used a 7-point scale.

**Analytic strategy.** We used mixed-effects models, with country included as a random effect. Each model included fixed effects for fusion to the

ingroup and the main effect and interaction terms for moderators. Continuous independent variables in moderation analyses were again scaled to facilitate interpretability. We calculated unstandardized regression coefficients for each model term, and marginal and conditional $R^2$ for the overall model, to act as effect size estimates. Due to the small amount of missing data ( < 5%) and concerns regarding the non-normality of residuals, all models were repeated using multiple imputation and bootstrapping. Nevertheless, these adjustments had a negligible effect on results.

## Reporting summary

Further information on research design is available in the Nature Portfolio Reporting Summary linked to this article.

## Results
### Study one

93% of all participants were fused with at least one group, with the following fusion rates to religion (74%), regional group (77%), political party (40%). Trust of outgroups was lowest in the political party dyad ($M$ = 5.11, $SD$ = 1.38) and highest in the religious dyad ($M$ = 5.45, $SD$ = 1.20), while general outgroup perceptions were lowest in the Lumad-Christian Settler dyad ($M$ = 64.65, $SD$ = 22.66) and highest in the Moro-Lumad dyad ($M$ = 71.10, $SD$ = 22.16). See Table S2 for a comparison of fused versus non-fused participants on all key variables. Zero-order correlations of all key variables are presented in Fig. 2.

Initial analyses found that participants who were fused to their regional ingroup ($M$ = 6.63) were significantly more open to cooperation versus those who were not fused ($M$ = 6.33), $t(270.22) = -4.73$, $p < 0.001$, $Cohen's$ $d = -0.43$, $95\% CI = -0.59$ to $-0.27$. Moderation analyses found that general outgroup perceptions consistently significantly moderated the relationship between fusion and outgroup trust (see Fig. 3; see Table 1). Across all intergroup dyads, the positive relationship between fusion and outgroup trust was stronger when outgroups were perceived more positively (i.e., 1 SD above the mean), while the relationship between fusion and outgroup trust was either null or negative when outgroups were perceived more coldly (i.e., 1 SD below the mean). The only exception was Model 5 (i.e., the political party dyad) in which fusion still had a positive effect on trust when the outgroup was perceived more coldly, albeit a much weaker effect than when the outgroup was perceived positively. The degree to which the outgroup was explicitly perceived as a threat was not a consistent moderator, only reaching significance in two out of five models (see Supplementary Table 3).

### Study two

Descriptive statistics and the zero-order correlations of all key variables are presented in Table 2 (see Table S6 for the descriptive statistics by country). Identity fusion was weakly positively correlated with willingness to cooperate and a perception that the ingroup would benefit from working with the outgroup.

Replicating Study 1, we found that general outgroup perceptions moderated the effect of fusion on willingness to cooperate with the outgroup (see Fig. 3; see Table 3). Simple slopes analyses indicated that fusion had a positive effect on willingness to cooperate when outgroups were perceived more positively (i.e., 1 SD above the mean; $t(1508) = 3.09$, $B = 0.18$, $p = 0.002$, 95% CI [0.07, 0.30]), but there was no statistically significant effect when groups were perceived more negatively (i.e., 1 SD below the mean; $t(1511) = -0.13$, $B = -0.01$, $p = 0.894$, 95% CI [−0.12, 0.11]). Moreover, perceptions that the ingroup has historically suffered from the behavior of the outgroup also moderated the relationship. While there was no statistically significant effect of fusion on willingness to cooperate when participants felt the outgroup was an historical threat (i.e., 1 SD above the mean; $t(1528) = 0.31$, $B = 0.02$, $p = 0.754$, 95% CI [−0.11, 0.15]), it had a positive effect when no such perception was present (i.e., 1 SD below the mean; $t(1528) = 3.67$, $B = 0.23$, $p < 0.001$, 95% CI [0.11, 0.35]). Finally, perceiving the outgroup as an explicit threat was not a statistically significant moderator.

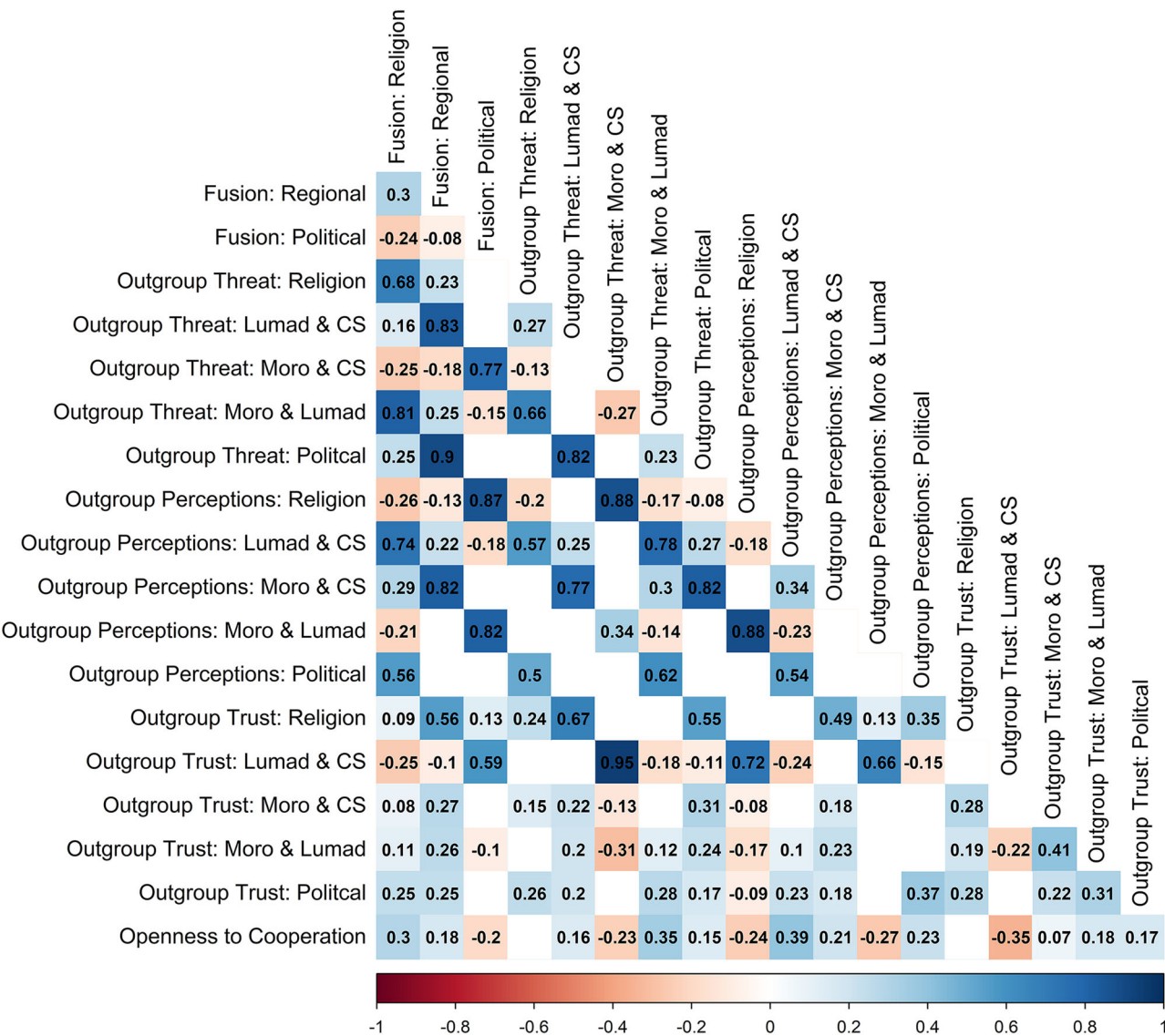

**Fig. 2 | Zero-order correlations for key variables.** CS is the abbreviation of Christian Settler. The reference group for the fusion term in all models was non-fused group members. Only correlations significant at the 0.05 level are displayed.

Darker red implies a stronger negative correlation and darker blue implies a stronger positive correlation.

Further analyses found that the effect of fusion on willingness to cooperate was moderated by perceived benefit to the ingroup (see Table 3; see Fig. 4). Simple slopes demonstrated that fusion had a positive effect on willingness to cooperate when ingroup benefits were seen as likely (i.e., 1 SD above the mean; $t(1528) = 3.33$, $B = 0.18$, $p < 0.01$, 95% CI [0.07, 0.28]), but there was no statistically significant effect when they were seen as unlikely (i.e., 1 SD below the mean; $t(1529) = -1.72$, $B = -0.09$, $p = 0.09$, 95% CI [−0.18, 0.01]). The effect of fusion on willingness to cooperate was also moderated by whether the intergroup relationship was perceived as zero-sum (i.e., when the outgroup gains, the ingroup loses). There was no statistically significant effect when of fusion on willingness to cooperate when the relationship was perceived as zero-sum (i.e., 1 SD above the mean; $t(1521) = -0.16$, $B = -0.01$, $p = 0.87$, 95% CI [−0.14, 0.12]), but a positive effect emerged when the relationship was not seen as zero-sum (i.e., 1 SD below the mean; $t(1523) = 4.24$, $B = 0.26$, $p < 0.01$, 95% CI [0.14, 0.38]).

Analyses including identification as an additional moderator found that the identity fusion and identification had different interactive patterns (see Table S7). Identification significantly interacted with negatively framed moderators (e.g., zero-sum, outgroup threat, historical threat), identity fusion significantly interacted with the positively framed moderator

(ingroup benefit), and there were no significant interactions when the moderator covered both a negative and positive frame in its anchors (outgroup perceptions).

## Discussion

The present paper explored how identity fusion influences intergroup trust and willingness to cooperate in complex social ecosystems. Specifically, we tested the *fusion-secure base hypothesis* which argues that fusion helps foster intergroup relations when outgroups are perceived favorably, or at least as non-threatening[1]. Study 1 supported this contention in an ecologically-valid setting, such that fusion positively predicted an openness to cooperate generally, and trust of a specific outgroup when it was perceived more positively. This finding was replicated across multiple intergroup dyads at various levels of abstraction (i.e., religion, regional group, and political party), while the moderating effect of perceiving the outgroup as an explicit threat was inconsistent. Study 2 replicated and expanded our model, finding that fusion positively predicted willingness to cooperate when the outgroup was perceived positively and not as a historic threat, but also when cooperation itself was viewed as beneficial and not zero-sum. Additional exploratory analyses that included identification as a moderator suggested that

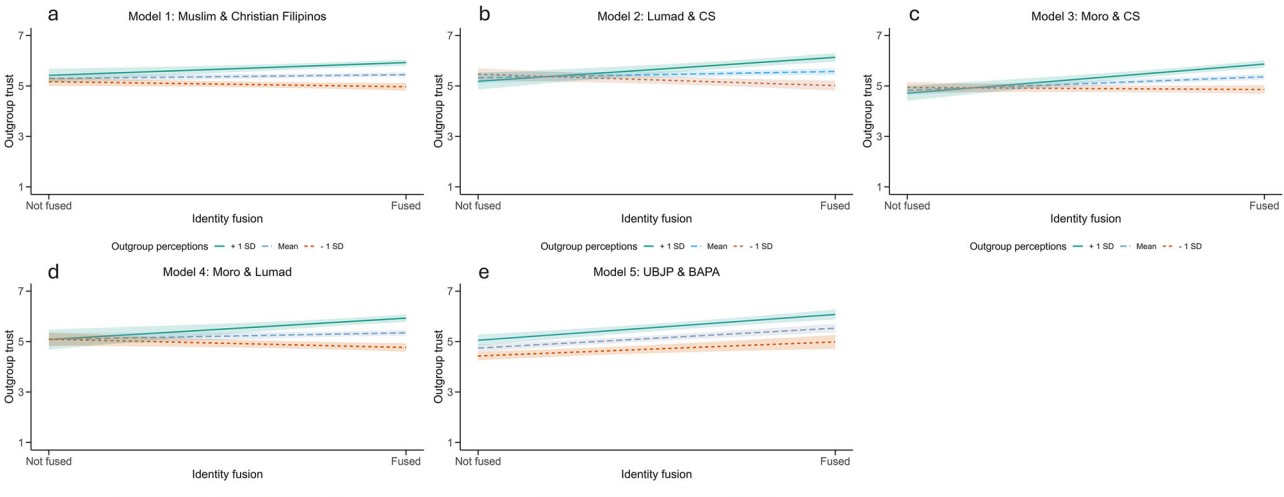

**Fig. 3 | The relationship between identity fusion and outgroup trust moderated by outgroup perceptions, controlling for group membership, in Study 1.** Models of the relationship between fusion and outgroup trust, as moderated by outgroup perceptions, for the following intergroup dyads: (**a**) Muslim & Christian Filipinos (*n* = 785 participants), (**b**) Lumad & Christian Settler (*n* = 428 participants), (**c**) Moro & Christian Settler (*n* = 608 participants), (**d**) Moro & Lumad (*n* = 594 participants), and (**e**) UBJP & BAPA (*n* = 535 participants). Confidence intervals are denoted by the shaded areas that protrude from the regression lines. Outgroup trust (y-axis) ranged from 1 to 7 and identity fusion (x-axis) was binary.

that fusion significantly interacted with positively framed moderators, and identification with negatively framed moderators. Overall, these results are generally consistent with the fusion-secure base hypothesis and suggest that fusion can have positive implications for intergroup relationships, albeit if the outgroup is perceived positively and the benefits of cooperation are salient.

These results reinforce those from other studies suggesting that identity fusion can engender positive intergroup relations under conducive conditions [20–22], offering a counterbalance to the field's overwhelming focus on intergroup violence. They also are in line with research from the social identity approach demonstrating that the social context can influence the intergroup behavior of strongly identified people[17], suggesting that motivations arising from identity fusion are subject to the same contextual cues. Importantly, this reinforces the original contention of the fusion-secure base hypothesis that identity fusion can promote either intergroup cooperation or hostility, depending on the intergroup context[1]. In the same way that a secure base might encourage someone to fight off a threat to the ingroup, it could empower them to overcome intergroup anxiety and trust or cooperate with a member of an outgroup.

It is worth considering why our scales explicitly measuring threat did not moderate the relationship as expected. One possibility is that the more overtly worded threat items were too blunt a tool to perform as intended. In both studies, but especially Study 1, the threat measures were very skewed towards perceiving the outgroup as non-threatening. While good news for the BARMM – perhaps reflecting decreased regional tensions[36] – this meant that these scales may have only captured extremely negative outgroup attitudes and missed more subtle gradations in outgroup perceptions. Moreover, explicit threat scales can only capture the negative side of outgroup perceptions (i.e., no threat vs. threat), whereas more general outgroup measures assess the full spectrum of attitudes (e.g., hostile vs. friendly) and so can better detect the relevant inflection point at which fusion begins to predict a willingness to cooperate. It is also worth noting that perceptions the outgroup had historically mistreated the ingroup (i.e., represented a past threat) did significantly moderate the relationship in Study 2, perhaps because it represents a more socially desirable threat measure.

More interestingly, these results may suggest that the original fusion-secure base hypothesis, which emphasized outgroup threat as the prime moderator, is overly simplistic. This possibility is reinforced by the clear moderating effect of the potential harms and benefits of cooperation itself in Study 2, highlighting that the nature of a specific relationship may matter as much or more than whether the outgroup is generally seen as threatening. History is littered with examples of sworn enemies working together for the greater good, with the momentary benefits of cooperation seemingly outweighing the outgroup's threat-status. Nevertheless, given that perceptions of the outgroup as a threat and viable cooperative partner are probably closely associated, this distinction may not always be important.

## Strengths
There are several strengths associated with the present study. Firstly, Study 1 examined intergroup relationships in a social ecosystem in which they are extremely salient and historically strained. In fact, the BARMM experienced a bloody Jihadist insurrection as recently as 2017, which displaced over 200,000 residents[37]. More broadly, the BARMM represents a context in which strong ingroup commitments and outgroup perceptions are meaningful, lending the data greater external validity. The meaningfulness of these commitments can be seen in the relatively high levels of identity fusion reported by participants, mirroring the fusion rates of other studies that have similarly focused on core social groups[10,38]. A further strength was our efforts to replicate our results, using varying measures and samples. The general moderating effect of outgroup attitudes was replicated across multiple group dyads from different levels of abstraction in Study 1, and then across multiple countries in Study 2, buttressing the robustness of our results. Finally, we were able to collect a relatively large sample over a broad geographic area, with a particular emphasis in Study 1 on reaching remote parts of the BARMM. This helped ensure that the various members of different intergroup relationships were adequately sampled.

## Limitations
Nonetheless, our results are not without limitations. First, the salience of the ongoing regional peace process in the BARMM could have made participants uncomfortable answering questions about outgroups in Study 1, particularly the explicit threat items, resulting in a tendency towards socially desirable responses. This was likely further exacerbated by the face-to-face administration of the survey, which was necessary given our extensive sampling strategy which targeted remote communities[27]. Steps were hence undertaken to address these concerns, including the use of more socially desirable and generic outgroup perception measures (i.e., whether the group was perceived coldly vs. warmly), matching enumerators with participants of the same ethnicity whenever possible, and not allowing enumerators to see participant responses. In any case, the replication of the effects in Study

**Table 1 | The relationship between identity fusion and outgroup trust moderated by outgroup perceptions, controlling for group membership**

| Terms | f | p | Adj. R² | df | n | B | CIs | t | p |
|---|---|---|---|---|---|---|---|---|---|
| Model 1: Muslim & Christian Filipinos | 34.02 | <0.001 | 0.14 | 780 | 785 | | | | |
| Intercept | | | | | | 5.62 | [5.43, 5.81] | 56.93 | <0.001 |
| Fusion to religion | | | | | | 0.15 | [−0.04, 0.34] | 1.54 | 0.120 |
| Religious group [Reference = Christians] | | | | | | −0.49 | [−0.66, −0.33] | −5.84 | <0.001 |
| Perceptions | | | | | | 0.13 | [−0.01, 0.26] | 1.80 | 0.07 |
| Fusion*Perceptions | | | | | | 0.35 | [0.18, 0.52] | 4.08 | <0.001 |
| Model 2: Lumad & CS | 18.73 | <0.001 | 0.14 | 423 | 423 | | | | |
| Intercept | | | | | | 5.4 | [5.19, 5.62] | 49.1 | <0.001 |
| Fusion to regional group | | | | | | 0.25 | [−0.01, 0.51] | 1.88 | 0.061 |
| Regional group [Reference = CS] | | | | | | −0.16 | [−0.39, 0.06] | −1.43 | 0.153 |
| Perceptions | | | | | | −0.14 | [−0.32, 0.04] | −1.50 | 0.133 |
| Fusion*Perceptions | | | | | | 0.70 | [0.47, 0.92] | 6.09 | <0.001 |
| Model 3: Moro & CS | 34.79 | <0.001 | 0.18 | 603 | 608 | | | | |
| Intercept | | | | | | 5.32 | [5.1, 5.53] | 49.08 | <0.001 |
| Fusion to regional group | | | | | | 0.54 | [0.31, 0.76] | 4.66 | <0.001 |
| Regional group [Reference = CS] | | | | | | −0.77 | [−0.97, −0.57] | −7.57 | <0.001 |
| Perceptions | | | | | | −0.11 | [−0.28, 0.05] | −1.34 | 0.182 |
| Fusion*Perceptions | | | | | | 0.62 | [0.42, 0.82] | 5.99 | <0.001 |
| Model 4: Moro & Lumad | 44.13 | <0.001 | 0.23 | 589 | 694 | | | | |
| Intercept | | | | | | 5.6 | [5.29, 5.91] | 35.51 | <0.001 |
| Fusion to regional group | | | | | | 0.26 | [-0.03, 0.55] | 1.78 | 0.076 |
| Regional group [Reference = Lumad] | | | | | | −0.79 | [−1, -0.58] | −7.47 | <0.001 |
| Perceptions | | | | | | −0.01 | [−0.21, 0.2] | −0.07 | 0.945 |
| Fusion*Perceptions | | | | | | 0.59 | [0.35, 0.82] | 4.93 | <0.001 |
| Model 5: UBJP & BAPA | 39.67 | <0.001 | 0.22 | 530 | 535 | | | | |
| Intercept | | | | | | 5.29 | [4.97, 5.61] | 32.55 | <0.001 |
| Fusion to political group | | | | | | 0.79 | [0.57, 1.01] | 7.08 | <0.001 |
| Political group [Reference = BAPA] | | | | | | −0.62 | [−0.94, −0.3] | −3.76 | <0.001 |
| Perceptions | | | | | | 0.31 | [0.17, 0.45] | 4.34 | <0.001 |
| Fusion*Perceptions | | | | | | 0.23 | [0.01, 0.45] | 2.08 | 0.038 |

CS is the abbreviation of Christian Settler. The reference group for the fusion term in all models was non-fused group members. Significant values at the 0.05 level are bolded.

2 suggests that the results of Study 1 were not overly biased by this or any other idiosyncratic feature of the BARMM context. A second limitation is the correlational nature of the results, which precludes drawing causal conclusions. However, it should be noted that a primary aim of the present study was to test the theory in an ecologically-valid, cross-cultural context, which lent itself more naturally to an observational design. It is encouraging that the present results reflect those from experimental studies that rely on more contrived manipulations[22]. Finally, the dependent variables (i.e., trust and willingness to cooperate) may have some conceptual overlap with some moderators (i.e., outgroup perceptions), although the moderate correlation between them suggests that they are likely measuring distinct constructs.

**Implications**

These findings have practical implications that can be explored further in future research. First, the notion that fusion promotes trust and willingness to cooperate has implications for whether *defusion* (i.e., the reversal of identity fusion) is generally an appropriate intergroup conflict intervention. Given that defusion is tantamount to removing a person's secure base, these results suggest it could have negative implications for intergroup relations. However, this needs to be reconciled with other research suggesting that former terrorists who distance themselves from their past comrades (i.e., defuse) endorse less extreme behavior[39]. It is likely that the norms of the group matter; terrorist groups have an explicit culture of violence and so defusion unsurprisingly reduces extremism, while, for groups not organized around an explicitly violent goal (e.g., religions), reshaping outgroup attitudes may be a more prudent intergroup conflict intervention. The present results support this assertion, highlighting the attractiveness of practical interventions like the *Twinning Project* which harness the prosocial effects of fusion with peaceful groups (i.e., local football clubs) to reduce recidivism among prisoners[40,41].

Future research should also work to disentangle the relative contributions of the different facets of outgroup perceptions to determine the core driver of the moderating effect. This could involve experimentation in which each facet is directly manipulated. Likewise, it would be interesting to examine the dynamics of identity fusion to multiple groups, as occurred in Study 1. The concept of fusion clusters[3] suggests that overlapping group

**Table 2 | Zero-order correlations and mean (standard deviation) for key variables**

|  | 1. | 2. | 3. | 4. | 5. | 6. | 7. | 8. |
|---|---|---|---|---|---|---|---|---|
| 1. Identity Fusion | 6.06 (1.00) | | | | | | | |
| 2. Willingness to Cooperate | 0.06* | 5.20 (1.83) | | | | | | |
| 3. Outgroup Threat | −0.04 | −0.24** | 3.28 (1.91) | | | | | |
| 4. Outgroup Perceptions | 0.02 | 0.49** | −0.29** | 4.57 (1.19) | | | | |
| 5. Historical Threat | −0.05 | −0.10** | 0.54** | −0.13** | 3.94 (1.97) | | | |
| 6. Ingroup Benefit | 0.13** | 0.51** | −0.14** | 0.23** | −0.22** | 4.81 (2.05) | | |
| 7. Zero-Sum Perceptions | −0.02 | −0.26** | 0.71** | −0.35** | 0.46** | −0.08** | 3.22 (1.93) | |
| 8. Identification | 0.61** | 0.13** | −0.08** | 0.11** | −0.07** | 0.14** | −0.09** | 6.10 (0.93) |

**Correlation is significant at the 0.01 level (2-tailed).
*Correlation is significant and the 0.05 level (2-tailed).

**Table 3 | The relationship between identity fusion and willingness to cooperate moderated by outgroup threat, outgroup perceptions, historical threat, ingroup benefit, and zero-sum perceptions**

| Terms | Marginal $R^2$ | Conditional $R^2$ | n | df | B | CIs | t | p |
|---|---|---|---|---|---|---|---|---|
| Model 1: Threat | 0.08 | 0.12 | 1529 | | | | | |
| **Intercept** | | | | 3.07 | 5.24 | [4.82, 5.67] | 27.21 | <0.001 |
| **Identity fusion** | | | | 1524.95 | 0.11 | [0.03, 0.2] | 2.54 | 0.011 |
| **Threat** | | | | 1524.85 | −0.50 | [−0.59, −0.41] | −11.05 | <0.001 |
| Fusion*Threat | | | | 1523.71 | −0.06 | [−0.15, 0.03] | −1.40 | 0.162 |
| Model 2: Perceptions | 0.23 | 0.25 | 1515 | | | | | |
| **Intercept** | | | | 2.81 | 5.22 | [4.96, 5.47] | 45.03 | <0.001 |
| **Identity fusion** | | | | 1500.14 | 0.09 | [0.01, 0.17] | 2.13 | 0.033 |
| **Perceptions** | | | | 1503.61 | 0.85 | [0.77, 0.93] | 20.45 | <0.001 |
| **Fusion* Perceptions** | | | | 1509.29 | 0.10 | [0.01, 0.18] | 2.28 | 0.023 |
| Model 3: Hist. Threat | 0.03 | 0.05 | 1532 | | | | | |
| **Intercept** | | | | 3.11 | 5.24 | [4.89, 5.59] | 32.85 | <0.001 |
| **Identity fusion** | | | | 1525.93 | 0.13 | [0.03, 0.22] | 2.72 | 0.007 |
| **Hist. Threat** | | | | 1486.36 | −0.23 | [−0.32, −0.13] | -4.76 | <0.001 |
| **Fusion*Hist. Threat** | | | | 1527.84 | −0.10 | [−0.19, −0.01] | −2.27 | 0.023 |
| Model 4: Benefit | 0.37 | 0.60 | 1535 | | | | | |
| **Intercept** | | | | 2.97 | 5.43 | [4.26, 6.59] | 10.25 | 0.002 |
| Identity fusion | | | | 1528.33 | 0.04 | [−0.03, 0.12] | 1.19 | 0.234 |
| **Benefit** | | | | 1527.57 | 1.35 | [1.26, 1.44] | 29.53 | <0.001 |
| **Fusion*Benefit** | | | | 1528.52 | 0.13 | [0.06, 0.2] | 3.65 | <0.001 |
| Model 5: Zero-Sum | 0.10 | 0.15 | 1527 | | | | | |
| **Intercept** | | | | 3.04 | 5.24 | [4.8, 5.67] | 26.65 | <0.001 |
| **Identity fusion** | | | | 1523.00 | 0.12 | [0.04, 0.21] | 2.82 | 0.005 |
| **Zero-sum** | | | | 1522.65 | −0.54 | [−0.63, −0.45] | −12.05 | <0.001 |
| **Fusion*Zero-sum** | | | | 1521.89 | −0.14 | [−0.23, −0.05] | −2.92 | 0.004 |

Significant values at the 0.05 level are bolded.

identities (e.g., fusion to religion, regional group, and political party) are mutually reinforcing and interact to motivate behavior, possibly augmenting the secure base effect. Further research is required to unpack the differing moderation patterns found when identification was introduced to the model. It is possible that people high in identification are more sensitive to threats (e.g., an avoidance orientation), whereas people high in fusion are more sensitive to perceived benefits (e.g., an approach orientation), although this is speculative and requires further investigation.

## Conclusion

Overall, our findings contribute to an understanding of how identity fusion, a form of extreme ingroup commitment, influences intergroup

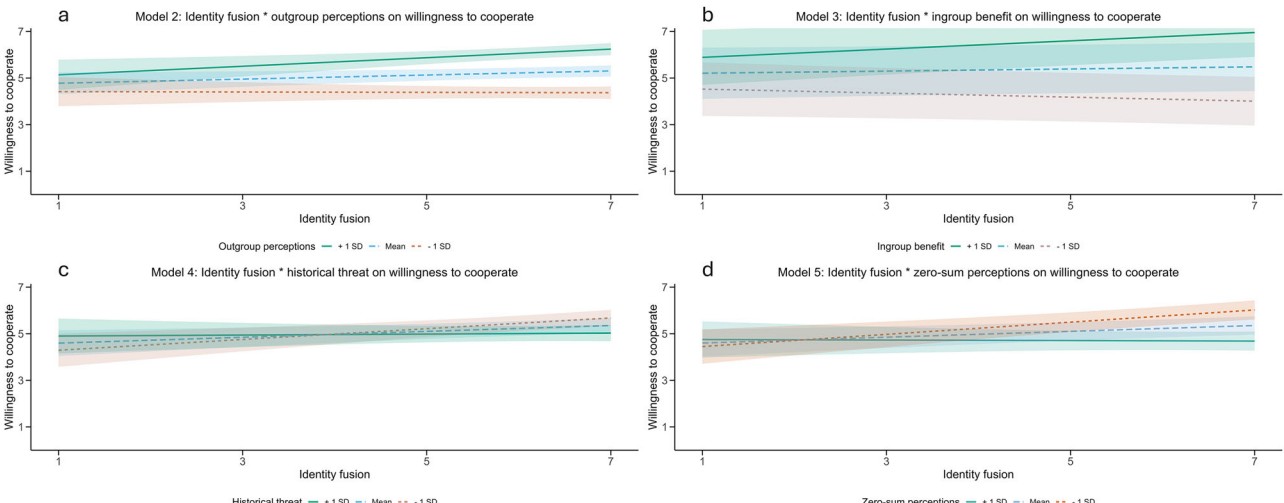

**Fig. 4 | The relationship between identity fusion and willingness to cooperate moderated by outgroup perceptions, ingroup benefit, historical threat, and zero-sum perceptions in Study 2.** Models of the interactions between identity fusion and (**a**) outgroup perceptions ($n = 1515$ participants), (**b**) ingroup benefit ($n = 1535$ participants), (**c**) historical threat ($n = 1532$ participants), and (**d**) zero-sum perceptions ($n = 1527$ participants). The interaction term in Model 1 (i.e., threat as the moderator) was not significant and so is excluded from the figure. Confidence intervals are denoted by the shaded areas that protrude from the regression lines. All scales ranged from 1 to 7.

relations in a complex social ecosystem. Results suggest that fusion increases a willingness to trust and cooperate with outgroups, albeit if perceptions of the outgroup and expected benefits are sufficiently favorable. These results support the fusion-secure base hypothesis and suggest that strong ingroup commitments play an important role in intergroup conflict and cooperation alike.

## Data availability
All data and questionnaires are available on the OSF at: https://osf.io/h7svn/?view_only=960ebac75dba4e98987a1693568e9554.

## Code availability
R version 4.3.1 was used for analyses. All code is available on the OSF at: https://osf.io/h7svn/?view_only=960ebac75dba4e98987a1693568e9554.

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

## Acknowledgements

This research was supported by an Advanced Grant from the European Research Council (ERC) under the European Union's Horizon 2020 Research and Innovation Programme (grant agreement No. 694986), and the Freedom of Religion or Belief Leadership Network Foreign (FoRBLN) grant from the UK's Foreign, Commonwealth & Development Office. The funders had no role in study design, data collection and analysis, decision to publish or preparation of the manuscript. We acknowledge the following people for their assistance gathering data: Christine Mbabazi Mpyangu, Sadiq Hussain, Haddy Njie, and Audax Mabulla.

## Author contributions

Jack W Klein: Contributed to conceptualization, methodology, formal analysis, investigation, data curation, writing- original draft preparation, visualization, project administration. Brock Bastian: Contributed to conceptualization, methodology, investigation, writing - review & editing, supervision. Emmanuel N. Odjidja: Contributed to methodology, investigation, resources, writing - review & editing, project administration. Samhita S. Ayaluri: Contributed to methodology, project administration. Christopher Kavanagh: Contributed to methodology, writing - review & editing. Alimudin M. Mala: Contributed to methodology, resources, project administration. Harvey Whitehouse: Contributed to conceptualization, methodology, investigation, writing - review & editing, supervision.

## Competing interests

The authors declare no competing interests.
