## [Transparent Peer Review file · Communications Psychology]

Identity fusion can foster intergroup trust and willingness to cooperate

Corresponding Author: Professor Jack Klein

Version 0:

Decision Letter:

Dear Professor Klein,

Thank you very much for your patience during the peer-review process. Your manuscript titled "Identity fusion can foster intergroup trust and cooperation" has now been seen by 2 reviewers, and I include their comments at the end of this message. They find your work of interest but raised some important points. We are interested in the possibility of publishing your study in *Communications Psychology*, but would like to consider your responses to these concerns and assess a revised manuscript before we make a final decision on publication.

We therefore invite you to revise and resubmit your manuscript, along with a point-by-point response to the reviewers. Please highlight all changes in the manuscript text file.

Editorially, we highlight the reviewers' key concerns regarding the theoretical relationship between the identity fusion and social identity. The methodological concerns (e.g., validity of identity fusion measures) should be addressed, and points that cannot be remedied via additional analysis or evidence should be thoroughly discussed as limitations.

In addition, we welcome the existence of preregistration, and ask you to ensure that your revision complies with our respective guidelines to facilitate future steps. Please report all deviations from the preregistration in the manuscript. Detailed documentation may be included in the Supplement. All originally preregistered hypotheses and analyses should be included, unless scientifically unsound, in which case the deviation needs to be highlighted and explained. Additional (exploratory) analyses may be included, but need to be labelled as post-hoc, non-preregistered. The full policy is here: <https://www.nature.com/commmspsychol/editorial-policies/preregistration-policy>

Please ensure you follow our statistical guidelines when reporting statistics (<https://www.nature.com/commmspsychol/submit/submission-guidelines#statistical-guidelines>). Please note in particular our requirements for the reporting and interpretation of null-results. Non-significant findings derived from null-hypotheses significance tests should be reported in full, but may not be interpreted. Where you interpret null results, this interpretation must be based on Bayes Factors or equivalence tests.

I am attaching an Editorial Requests Table that details critical reporting requirements for the revised manuscript. Please attend to each item and ensure your manuscript is fully compliant. If your revised manuscript is not aligned with these requests on major issues, such as those concerning statistics, it may be returned to you for further revisions without re-review.

Please submit the following items:

- Revised manuscript

- Point-by-point response to the referees' comments
- Cover letter (as a separate document)
- [Nature Research Reporting Summary](https://www.nature.com/documents/nr-reporting-summary.zip)
- [Editorial Policy Checklist](https://www.nature.com/documents/nr-editorial-policy-checklist.pdf)
- Completed Editorial Request Table (attached).

via this link: Link Redacted .

Additional guidance is available in our style and formatting guide [Communications Psychology formatting guide](https://www.nature.com/documents/commpsychol-style-formatting-guide-accept.pdf).

Best regards,

Hannah Nam

Hannah Nam, PhD
 Editorial Board Member
 Communications Psychology
 orcid.org/0000-0003-3006-0584

REVIEWER EXPERTISE:

Reviewer #1: social identity, identity fusion, intergroup attitudes

Reviewer #2: social identity, interethnic conflict, intergroup attitudes

REVIEWER REPORTS:

Reviewer #1 (Remarks to the Author):

The paper "Identity fusion can foster intergroup trust and cooperation" reports two empirical studies on the relationship between identity fusion and the positive intergroup outcomes trust and willingness to cooperate in social environments in which intergroup tensions have historically been present. The paper is concise and well written, and I enjoyed reading it. It makes a potential contribution to the literature by examining when identity fusion may relate to positive intergroup relations rather than the usual focus on extreme and negative behaviours (e.g., terrorism, self-sacrificing). I would also like to highlight that samples were obtained from communities that are largely understudied in a way that allowed participants in remote communities to participate (Study 1) and through paper and pencil rather than only through online research platforms across several non-Western countries (Study 2), which is impressive to see.

There are some aspects of the paper that I believe can be improved and some questions of a theoretical nature that need to be addressed, in my opinion, for this paper to make a full contribution to the literature:

Validity of the conclusion: My main concern relates to the strength of the conclusion, that is, whether the effects are theoretically consistent with an identity fusion explanation. There are several reasons why it is difficult to exclude social identification as the more parsimonious explanation and I would like to see this discussed in the paper before drawing conclusions about identity fusion:

1) Social identification was not statistically controlled for in the study despite its overlap with the concept of identity fusion

(also in measurement terms) and longstanding research that shows that higher levels of identification are related to more outgroup trust and liking when relationships are collaborative rather than competitive (e.g., Montoya & Pittinsky, 2011).

2) Given that identity fusion has been theorised as quite an extreme form of identity, e.g., making self-sacrifice and extremist/terrorist behaviour more likely, it strikes me as problematic that extremely high rates of identity fusion (93%) are found in Study 1 (fusion rates are not reported for Study 2). To me, it seems unlikely that the vast majority of a population holds an extreme self-concept. In contrast, it would indeed be expected that people highly identify with several relevant social identities within their social environment.

3) Similarly, the study reported that it is not uncommon in the sample that multiple identities are fused to the personal self. How would a person manage such a self-concept, and what would be the consequences? Would behaviour towards the various outgroups be identical since, based on their self-concept, the same set of values and beliefs inform their actions (see low identity complexity)? A social identity explanation would have an easier time with multiple identities thanks to the concept of salience.

Measurements:

1) Identity fusion: Have identity fusion measures been validated in countries with a more interdependent rather than independent culture such as the communities included in the studies? I wonder whether the pictorial measure in particular may be vulnerable to participants simply seeing themselves as interdependent with their communities rather than showing identity fusion when encircling self within a community.

2) Cooperation: As willingness to cooperate was measured rather than actual cooperation, it would be better to have the title and narrative of the paper reflect this.

Analyses:

1) Model 5: this is likely to be a simple reporting issue, but it says that "Fusion to regional group" is added as a main effect rather than "Fusion to political group" (or perhaps this is why Model 5 shows inconsistent results?)

2) Aggregating across social groups: As the regression analyses in Study 1 show, the status of the group (e.g., majority v minority) affects trust directly. Most theories of intergroup relations account for the finding that high- and low-status groups perceive the world and act within it rather differently. However, the social environment that is highlighted in the introduction is not currently reflected in the analyses. Unfortunately, the reviewing files did not appear to include the supplementary materials mentioned in the document (and I could not open it on OSF either), so I have too little information on sample demographics (only gender for Study 1 and country for Study 2) to judge whether the sample consisted of equal numbers of high-status (e.g., Moro) and low-status group members (e.g., Christian settlers) - and it may therefore be feasible to analyse the data by groups - or whether results are dominated by majority group participants. Similarly, in Study 2, participants were aggregated across countries and irrespective of whether they are part of a religious majority or minority within their country. I wonder whether separate analyses would yield stronger, more interpretable findings.

3) Please also provide Study 1 correlations as you do for Study 2

4) Figures: y-axes were not consistent with the full scale and missing for Study 2

Thank you for the opportunity to review this fascinating paper and I hope my comments come across as a constructive contribution to strengthen the conclusions of the paper.

Miriam Koschate

Reviewer #2 (Remarks to the Author):

Review Comments

■ Limitations of Measurement Methods and Design

A major concern is that both the dependent variables—"outgroup trust" and "outgroup cooperation"—and the moderator "perception" appear to capture aspects of outgroup positivity, which may lead to some conceptual redundancy. This raises the possibility that the interaction effects may partly reflect measurement overlap rather than distinct psychological processes. Although the analyses demonstrate that several variables act as moderators, it remains unclear which specific facet of outgroup positivity is the core driver of the moderating effect. Future research should aim to more clearly separate these constructs and provide robust empirical evidence and theoretical mechanisms to pinpoint the primary contributor.

■ Theoretical Background and Hypothesis Development

The manuscript is intriguing in its suggestion—based on the "fusion-secure base hypothesis"—that a strong sense of ingroup affiliation can also foster trust and cooperation toward outgroups. This is an important and timely contribution to the literature. However, the discussion would benefit from a more detailed comparison with previous research on identity fusion as well as traditional social identity theory. Such discussion would help clarify the novelty and distinctiveness of the hypotheses. Given the well-documented link between identity fusion and pro-group extreme behavior (i.e., outgroup aggression), a deeper exploration of the underlying psychological processes that account for both violent and cooperative behavior would strengthen the theoretical framework. In particular, while the mechanism centered on threat is discussed, the absence of a threat moderation effect in Study 2—in contrast to the moderation observed for perception—warrants further careful interpretation and discussion.

■ Interpretation of Results and Consistency in Discussion

The findings from both Study 1 and Study 2 generally support the proposed hypotheses, which is highly commendable given the complexity of the topic and the large-scale, international data utilized. However, since the measures for "perception" and

“threat” regarding outgroups and the dependent variables “trust” and “cooperation” all assess aspects of outgroup positivity, there is potential overlap in what is being measured. This raises the possibility that the observed interaction effects might be influenced by shared measurement variance rather than distinct underlying processes. In particular, the fact that interaction effects emerged only for perception and not for threat suggests that the results might partly be driven by common measurement elements. This issue should be acknowledged and discussed to provide a more nuanced interpretation of the findings.

■ Overall Evaluation

Overall, this manuscript is an important contribution to the field, offering valuable insights into how identity fusion may promote positive intergroup relations using large-scale, cross-cultural data. The innovative approach of examining multiple interaction effects is particularly commendable. While there are areas for improvement—such as deepening the theoretical framework, refining measurement methods, and enhancing the rigor of statistical analyses—the study’s strengths and its potential to advance our understanding of complex intergroup dynamics are clear. Addressing the methodological challenges related to overlapping constructs will further enhance the impact and clarity of this promising work.

If you experience problems in linking your ORCID, please contact the Platform Support Helpdesk.

Version 1:

Decision Letter:

Dear Professor Klein,

Your manuscript titled "Identity fusion can foster intergroup trust and willingness to cooperate" has now been seen by our reviewers, whose comments appear below. In light of their advice I am delighted to say that we are happy, in principle, to publish a suitably revised version in Communications Psychology.

We therefore invite you to revise your paper one last time to address the remaining concerns of our reviewers and a list of editorial requests. At the same time we ask that you edit your manuscript to comply with our format requirements and to maximise the accessibility and therefore the impact of your work.

Editorially, we highlight the following necessary revisions:

In response to Reviewer #1, we ask that the presentation of the conceptual background and contribution is further improved. Related to their request regarding the additional analysis, please note the following:

- The additional analyses should be included either in the main manuscript or the Supplement.
- The Supplement must be a separate file, submitted with your revisions. Having a Supplement externally, e.g. on OSF does not suffice

Regarding the preregistration, please note that the presentation must align with our policy for preregistration, which includes the necessity to point out in the main manuscript any deviations from the preregistration and to include all preregistered

analyses, unless fundamentally flawed or unfeasible (which would need to be declared).

We ask you to also pay special attention to the requirements for an expanded ethics and inclusion statement, as detailed on the "Editorial Requests Table".

EDITORIAL REQUESTS:

Please outline your response to each request on the "Editorial Requests Table" in the right hand column. Please upload the completed table with your manuscript files as a Related Manuscript file.

SUBMISSION INFORMATION:

OPEN ACCESS:

* DATA AVAILABILITY:

Link Redacted

Best regards,

Marike

Marike Schiffer, PhD
Chief Editor
Communications Psychology

REVIEWERS' COMMENTS:

Reviewer #1 (Remarks to the Author):

I would like to thank the authors for their thorough engagement with the issues that I raised, particularly in the rebuttal letter, along with the additional analyses conducted. The following points are intended to encourage the authors to bring out this depth of theorising and analysis in the article as well.

R1.1: The differentiation from identification still appears to hinge on fusion as the predictor of extremism - particularly in the opening part of the paper where the only clarifying sentence is: "It is distinguished from other measures, such as identification from social identity theory, in part through its propensity to strongly predict extreme and violent pro-group behavior"; for me, this undermines the theoretical argument that identity fusion is a better predictor for trust and willingness to cooperate than social identification.

The authors later go into more depth on why identity fusion may act as a secure base. However, the assertion that social identification merely translates norms into behaviour but does not otherwise contribute to a sense of belonging with the group and group-based efficacy is not consistent with the social identity literature. Literature on social identity in organisations as well as on the social cure show that social identity provides a sense of belonging, purpose and meaning along with social support to group members and a sense of efficacy (e.g., review paper by Haslam et al., 2022; <https://doi.org/10.1016/j.copsyc.2021.07.013>). As the argument why identity fusion is a secure base relates to similar concepts, I would recommend that such overlap in theorising is correctly acknowledged and differences between constructs are made clear.

R1.2: I would like to see the analyses reported in the rebuttal letter which include social identification included in the article, particularly for Study 2. It seems misleading to me to assert in the article that identity fusion increases a willingness to cooperate when there is low historical threat or less zero-sum thinking, when this is not, in fact, the case once social identification is controlled for. In fact, the analyses provide an interesting avenue for future research by pointing towards an identity fusion/benefit interaction on willingness to cooperate rather than an identity fusion/threat interaction (as hypothesised). I could not find a pre-registration for Study 2, so it seems to me that Study 2 is largely exploratory in any case.

Interpretation: The rebuttal letter states that the "main purpose of our manuscript was to demonstrate that identity fusion can promote positive intergroup relations under certain conditions, rather than to show that social identification cannot". Perhaps I misunderstand the table in the rebuttal letter. To me, the re-analysis appears to show that ONLY identification has a POSITIVE main effect. In addition to the main effect, identification appears to act on willingness to cooperate in the way fusion was theorised to act, that is, when threat is low, whereas fusion only has an effect when benefits are perceived. This clarification is a valuable contribution to current theorising on identity fusion as a secure base but also social identification and willingness for intergroup cooperation.

R1.3: Thank you for the explanation on why high fusion rates may be expected in the specific social contexts studied here. Other readers may wonder about this too so it would be good to include a sentence on this in the discussion section of the article as well.

R1.5: Thank you for the background on the fusion measures. It would be good to include a sentence in the methods section on the validity of fusion clusters in interdependent cultures to reassure readers.

R1.6: Thank you for changing the title and variable in the paper. However, there are several instances where intergroup cooperation still needs to be replaced with willingness to cooperate to avoid confusion/misinterpretation.

R1.8: Is the dyad size table going to be included in supplementary analyses? I think it would be useful and strengthens the paper's interpretability.

R1.9: Thank you for including the correlations table. Standard APA correlation tables are easier to read, in my opinion, but I leave this to the editor/journal.

Other:

p.4 - increased adherence TO shared norms

p.13 – Zero-order correlations of all key variables are presented in Table 2 – should this be Table 1?

Reviewer #2 (Remarks to the Author):

Comments for Author

Thank you for your thoughtful and well-structured revisions. The concern regarding potential conceptual overlap between outgroup perception and the outcome variables has been clearly acknowledged and addressed. The authors provide a reasonable empirical justification, and the inclusion of this point in the limitations section strengthens the transparency of the work. The discussion of future directions—such as experimentally disentangling facets of perception—adds further value.

The expanded theoretical discussion also improves the manuscript. The contrast between identity fusion and social identity

theory is now clearer, and the nuanced interpretation of the null effect for threat offers a plausible and well-argued explanation. These additions make the overall argument more coherent and compelling.

Taken together, the revisions adequately address the issues raised and enhance both the clarity and theoretical contribution of the manuscript. Overall, the revisions are sufficient and the manuscript is now well-prepared for publication.

21-May-2025

Dear Reviewers,

Before turning to your comments, we would like to take this opportunity to thank you for reading our manuscript and providing such constructive feedback. We have made numerous changes to the manuscript and endeavored to address all concerns raised. Please note that the page numbers provided refer to the version of the manuscript with track changes.

Reviewer #1 (Remarks to the Author):

The paper “Identity fusion can foster intergroup trust and cooperation” reports two empirical studies on the relationship between identity fusion and the positive intergroup outcomes trust and willingness to cooperate in social environments in which intergroup tensions have historically been present. The paper is concise and well written, and I enjoyed reading it. It makes a potential contribution to the literature by examining when identity fusion may relate to positive intergroup relations rather than the usual focus on extreme and negative behaviours (e.g., terrorism, self-sacrificing). I would also like to highlight that samples were obtained from communities that are largely understudied in a way that allowed participants in remote communities to participate (Study 1) and through paper and pencil rather than only through online research platforms across several non-Western countries (Study 2), which is impressive to see.

There are some aspects of the paper that I believe can be improved and some questions of a theoretical nature that need to be addressed, in my opinion, for this paper to make a full contribution to the literature:

R1.1 *Validity of the conclusion: My main concern relates to the strength of the conclusion, that is, whether the effects are theoretically consistent with an identity fusion explanation. There are several reasons why it is difficult to exclude social identification as the more parsimonious explanation and I would like to see this discussed in the paper before drawing conclusions about identity fusion:*

Response:

Thank you for this very relevant comment. We address your more specific concerns in the comments below, but it is worth first noting that the premise for this paper comes from previous theoretical (Klein & Bastian, 2023) and empirical (Klein, Greenaway, & Bastian, 2024) work on identity fusion. This perspective suggests that the same features of identity fusion that empower ingroup members to engage in intergroup conflict – the capacity for identity fusion to foster a “secure base” – also motivates intergroup cooperation. Similar explanations are not present in social identity accounts (e.g., Montoya & Pittinsky, 2011). These emphasize how contextual cues direct intergroup behavior (e.g., introducing the outgroup in a cooperative vs. competitive context), rather than how a supportive ingroup allows group members to overcome intergroup anxiety and approach outgroups in the first place.

We contend that both ingroup motivation and contextual perceptions are important. As an analogy, identity fusion could be considered the “engine” that drives behavior while perceptions of the context are the “steering wheel” that directs it towards cooperation or conflict. When research from the social identity approach does consider “the engine”, it focuses on how highly identified people adhere to group norms and group-oriented behavior. While relevant, these motivations strike us as a less compelling explanation for intergroup “approach motivations” than our secure base explanation (we return to this idea in response to R1.2).

For example, past research has found that ingroup trust – a proxy for a secure base – mediates the effect of identity fusion on outgroup trust (Klein, Greenaway, & Bastian, 2024). We replicated this in Study 1, with ingroup trust clearly mediating the effect of identity fusion on intergroup trust. However, as we did not measure ingroup trust in Study 2, we opted not to include these analyses in the manuscript.

In any case, the current article serves as a correction to the idea that extreme forms of group alignment only contribute to intergroup conflict, which the identity fusion literature focuses on. Instead, we demonstrate how identity fusion can have positive intergroup outcomes in the right social context. Given that there is considerable evidence that identity fusion is a stronger predictor of support for violence than identification (Varmann, et al., 2024), showing how identity fusion can also promote cooperation represents a worthwhile contribution in its own right. Hence, the primary purpose of this study was to examine whether identity fusion could have a positive impact on intergroup relations in a dynamic, ecologically-valid setting.

We have now made the contrasting theoretical approach to the social identity approach clearer in the introduction (pg. 4):

“This differs from similar work based on the social identity perspective ¹⁶, which has argued that increased adherence shared norms and group-oriented behavior drive exploratory motivations in highly-identified people.”

And later in the discussion (pg. 21):

“These results reinforce those from other studies suggesting that identity fusion can engender positive intergroup relations under conducive conditions ¹⁷⁻¹⁹, offering a counterbalance to the field’s overwhelming focus on intergroup violence. They also are in line with research from the social identity approach demonstrating that the social context can influence the intergroup behavior of strongly identified people ¹⁶, suggesting that motivations arising from identity fusion are subject to the same contextual cues. Importantly, this reinforces the original contention of the fusion-secure base hypothesis that identity fusion can promote either intergroup cooperation or hostility, depending on the intergroup context ¹. In the same way that a secure base might encourage someone to fight off a threat to the ingroup, it could empower them to overcome intergroup anxiety and trust or cooperate with a member of an outgroup.”

We have also made the primary contribution of the paper – that identity fusion can positively influence intergroup relations in the right circumstances – more explicit (pg. 4-5):

“The present study seeks to test the moderation component of the fusion-secure base hypothesis in naturalistic settings, in which outgroup perceptions vary, thereby illustrating the circumstances under which identity fusion elicits trust and willingness to cooperate. In doing so we illustrate that identity fusion can have positive outcomes in the right social context, counterbalancing the prevailing focus of the identity fusion literature on its contribution to intergroup conflict.”

R1.2 *Social identification was not statistically controlled for in the study despite its overlap with the concept of identity fusion (also in measurement terms) and longstanding research that shows that higher levels of identification are related to more outgroup trust and liking when relationships are collaborative rather than competitive (e.g., Montoya & Pittinsky, 2011).*

Response:

Although the primary purpose of this study was to first establish the circumstances under which identity fusion could promote cooperation, we agree that it is also important to disentangle identity fusion’s effect from “mere” identification. Indeed, there are theoretical reasons to believe that identity fusion is a stronger motivator than social identification of intergroup cooperation – largely related to its capacity to empower ingroup members to overcome intergroup anxiety, whether to confront or cooperate with an outgroup (Klein & Bastian, 2023) – and subsequent research has found that identity fusion is a stronger and more consistent predictor than identification of outgroup trust, at least in benign settings (Klein, Greenaway, & Bastian, 2024; Vázquez et al., 2023).

To properly address your comment, we reanalyzed the regressions from both studies including identification measures as control variables. In Study 1 we had the single item measure “I identify with my group” (Postmes et al., 2011). In Study 2 we also had the Postmes et al., (2011) measure, as well as the following three items: (1) I have a lot in common with the ingroup, (2) I connect with the values of the ingroup, and (3) I feel a sense of belonging with the ingroup. Including the identity fusion and identification measures into an exploratory factor analysis found that all items loaded onto a “fusion” or “identification” factor, excluding “I feel a sense of belonging with the ingroup” which was subsequently dropped.

First, we rerun our regression models controlling for identification as a main effect. This had a negligible effect on results in both studies, and the identity fusion interactions remained significant.

Next, we included identification as an interaction term (i.e., identification * the relevant moderator in each regression). In Study 1 we found that both fusion and identification remained significant moderators in the regional identity models, although fusion was no longer significant in the religion and political models. Subsequent linear hypothesis tests suggested that the difference between the identification and fusion interaction coefficients was non-significant. Prior research has shown that the pictorial measure of fusion is a much poorer predictor of fusion outcomes than the verbal measure, often performing worse than

identification measures (Varmann, et al., 2024), so these results were not particularly surprising.

Including identification as an interaction effect in Study 2, which used an abbreviated version of the superior verbal fusion scale, yielded more interesting and interpretable results:

Terms	Marginal R²	Conditional R²	df	B	CI s	t	p
Model 1: Threat	0.1	0.15					
Intercept			3.07	5.22	[4.79, 5.66]	26.57	<.001
Identity fusion			1503.71	-0.05	[-0.16, 0.06]	-0.86	.389
Identification			1501.49	0.28	[0.17, 0.39]	4.98	<.001
Threat			1503.98	-0.48	[-0.57, -0.39]	-10.66	<.001
Fusion*Threat			1501.96	0.05	[-0.05, 0.16]	0.99	.323
ID*Threat			1501.36	-0.22	[-0.33, -0.11]	-3.89	<.001
Model 2: Perceptions	0.24	0.26					
Intercept			2.85	5.21	[4.94, 5.47]	43.28	<.001
Identity fusion			1489.65	0.01	[-0.09, 0.11]	0.12	.906
Identification			1488.92	0.15	[0.05, 0.25]	2.84	.005
Perceptions			1482.21	0.85	[0.77, 0.93]	20.29	<.001
Fusion* Perceptions			1488.44	0.05	[-0.05, 0.14]	0.98	.328
ID* Perceptions			1487.60	0.09	[-0.01, 0.18]	1.73	.083
Model 3: Hist. Threat	0.05	0.08					
Intercept			3.11	5.22	[4.86, 5.58]	32.04	<.001
Identity fusion			1506.90	-0.05	[-0.16, 0.07]	-0.82	.414
Identification			1504.83	0.30	[0.18, 0.41]	5.15	<.001
Hist. Threat			1469.59	-0.20	[-0.29, -0.11]	-4.23	<.001
Fusion*Hist. Threat			1506.31	< .01	[-0.11, 0.11]	0.02	.984
ID*Hist. Threat			1504.70	-0.21	[-0.32, -0.09]	-3.48	<.001
Model 4: Benefit	0.38	0.61					
Intercept			2.97	5.42	[4.25, 6.61]	10.1	.002
Identity fusion			1508.31	-0.02	[-0.11, 0.07]	-0.4	.687
Identification			1508.24	0.13	[0.04, 0.22]	2.76	.006
Benefit			1507.80	1.36	[1.27, 1.45]	29.42	<.001
Fusion*Benefit			1508.60	0.10	[0.02, 0.19]	2.32	.020
Identification*Benefit			1508.11	0.01	[-0.08, 0.1]	0.28	.783
Model 5: Zero-Sum	0.11	0.16					
Intercept			3.05	5.22	[4.79, 5.65]	26.63	<.001

Identity fusion	1503.49	< 0.01	[-0.11, 0.1]	-0.08	.936
Identification	1501.54	0.23	[0.12, 0.34]	4.15	<.001
Zero-sum	1503.82	-0.52	[-0.61, -0.43]	-11.48	<.001
Fusion*Zero-sum	1501.59	-0.07	[-0.19, 0.04]	-1.28	.200
ID*Zero-sum	1501.96	-0.13	[-0.24, -0.02]	-2.22	.027

Notes: Significant values at the .05 level are bolded.

In short, identification was significant when the moderator was negatively framed (e.g., zero-sum, threat), fusion was significant when the moderator was positively framed (benefit), and neither was significant when the moderator covered both a negative and positive frame in its anchors (perceptions). This pattern of results could suggest high levels of identification prompts people to quickly break-off cooperation when they perceive a threat, while high levels of fusion can compel people to pursue cooperation if they perceive it as beneficial (irrespective of potential threat). This conforms to our earlier discussion of identity fusion promoting a secure base and “approach” motivations.

While interesting, we are mindful that these analyses are essentially exploratory and should be interpreted cautiously. We have hence elected to include these analyses in the supplementary materials, but would be open to relocating them to the main manuscript if the reviewer felt it was necessary. Finally, it is worth reiterating that the main purpose of our manuscript was to demonstrate that identity fusion can promote positive intergroup relations under certain conditions, rather than to show that social identification cannot. We also feel that our ‘secure base’ theoretical approach offers a parsimonious explanation of the present findings that does not appear in other accounts.

R1.3 *Given that identity fusion has been theorised as quite an extreme form of identity, e.g., making self-sacrifice and extremist/terrorist behaviour more likely, it strikes me as problematic that extremely high rates of identity fusion (93%) are found in Study 1 (fusion rates are not reported for Study 2). To me, it seems unlikely that the vast majority of a population holds an extreme self-concept. In contrast, it would indeed be expected that people highly identify with several relevant social identities within their social environment.*

Response: While identity fusion has certainly been characterized associated with extremism, this is more in terms of the extreme outcomes it can produce (e.g., a willingness to fight threatening outgroups) than rarity. Despite the focus on extremism in the fusion literature, more common instances of fusion typically produce peaceful forms of loyalty and caring for ingroup members. For instance, fusion with one’s family is very common, and the lack of intergroup rivalries means it typically does not inspire extremism (although familial blood feuds are common in some cultural systems). The association between fusion and violent extremism is thought to be triggered by other variables, such as outgroup threat, violence-condoning norms, and demonization of enemies, and is hence not attributable to fusion alone (e.g., Ebner et al.,2022). In the present study we deliberately targeted ingroups that were especially meaningful to participants (e.g., religion, regional/ethnic group) and so the

relatively high fusion rates were not unexpected. Correspondingly, the relatively lower rate of fusion to political party (40%) suggests that this does not reflect an acquiescent response bias.

Previous studies have also found that identity fusion can be very common. Perhaps the most comparable study is Purzychi & Lang (2019) which used the pictorial measure to examine fusion to their community across eight cross-cultural populations. Identity fusion could be extremely high, with 85% of the hunter-gatherer Hazda people of Tanzania reporting fusion to their community (exceeding fusion to regional group – 77% – in the present study). Other research using the pictorial scale has found similar results; for example, Whitehouse et al., (2014) found that 99% of Libyan soldiers were fused with their own family, 97% with their battalion, and 96% with other battalions.

We do not report fusion rates for Study 2 because we used a continuous measure of identity fusion, which has better psychometric properties and is not typically dichotomized (see Gómez et al., 2011). The mean fusion scores were also high, although comparable to other studies examining fusion to religion (Wibisono et al., 2022).

R1.4 *Similarly, the study reported that it is not uncommon in the sample that multiple identities are fused to the personal self. How would a person manage such a self-concept, and what would be the consequences? Would behaviour towards the various outgroups be identical since, based on their self-concept, the same set of values and beliefs inform their actions (see low identity complexity)? A social identity explanation would have an easier time with multiple identities thanks to the concept of salience.*

Response: Thank you for this comment. We should make clear that the various identities examined in Study 1 have considerable overlap. For instance, almost all Moro are Muslim and Christian Settlers are Christian; even the political parties are strongly tied to religious and ethnic identities (e.g., the UBJP is explicitly Islamic and Moro).

The concept of “fusion clusters” is relevant, recently discussed in Swann, Klein, & Gómez (2024). Fusion clusters describe fusion to multiple targets, such as a jihadist who reports feeling fused to Islam, fellow jihadists, the prophet, and God, and are common in the fusion literature. For example, research examining Spanish support for Ukraine in its conflict with Russia found that fusion to the Ukrainian president Zelensky and fusion to freedom were significantly correlated (Gomez et al., 2023). We suspect that these identities are mutually reinforcing, and that fusion to one may even make fusion to others more likely. If the identities do not conflict, the consequence may be that a person is especially motivated to engage in pro-group behavior for all their fused identities. Fusion clusters hence explain how overlapping identities can interactively motivate behavior. By contrast, while the concept of salience in the social identity approach can predict when an individual would switch identities, it does not help explain why a strong group alignment might predict intergroup cooperation.

Nevertheless, to the extent that these identities diverge in their degree of fusion or in beliefs and values, we agree that salience plays a role. In these cases, intergroup relations are likely most influenced by the ingroup at the same level of abstraction as the outgroup. For example, willingness to trust another ethnic group is probably most influenced by fusion to one’s own

ethnic group, as the value of cooperation is made more salient. This is the reason we focused on regressions that compare group identities at the same level of abstraction. Below is an extract from Klein & Bastian (2023) that discusses this idea in more depth:

“Although the model should apply to any conceivable out-group member (e.g., a stranger and foreigner), fused actors may have a particularly strong rationale to cooperate when the in-group and out-group are at equivalent levels of abstraction (e.g., two communities and two nations). This is because “equivalent” groups are more likely to face similar problems and have similar goals, which makes the mutual benefits of cooperation more tangible. Moreover, equivalent groups will more likely provide an equal level of contribution to a joint enterprise; this is particularly relevant given that collective efficacy (i.e., the perceived ability of the groups to effectively work together) is predictive of cooperation (Klavina & van Zomeren, 2018). The heightened salience of shared goals and collective efficacy makes a cooperative intergroup relationship between equivalent groups an attractive prospect for an agentic fused actor—drawing upon the resources of their in-group as a secure base—to explore and act on.”

We have now made the overlapping nature of these identities clear when discussing the BARRM social context (pg. 8):

“These ingroup identities have considerable overlap; for example, almost all Moro are Muslim and Christian Settlers are Christian, with even UBJP and BAPA strongly tied to religious and ethnic identities.”

And speculated about the role of fusion clusters in the discussion (pg. 24-25):

“Likewise, it would be interesting to examine the dynamics of identity fusion to multiple groups, as occurred in Study 1. The concept of “fusion clusters”³ suggests that overlapping group identities (e.g., fusion to religion, regional group, and political party) are mutually reinforcing and interact to motivate behavior, possibly augmenting the secure base effect.”

Measurements:

R1.5 Identity fusion: *Have identity fusion measures been validated in countries with a more interdependent rather than independent culture such as the communities included in the studies? I wonder whether the pictorial measure in particular may be vulnerable to participants simply seeing themselves as interdependent with their communities rather than showing identity fusion when encircling self within a community.*

Response: It is certainly possible that identity fusion operates differently in different independent vs. interdependent cultures, and there have been attempts to validate identity fusion in independent and interdependent cultures. In fact, most of the studies on identity fusion to date, including those establishing the psychometric validity of the verbal measure of fusion (Gómez et al., 2011), have been conducted on Americans (an independent culture) and Spaniards (a relatively interdependent culture).

Fusion has also been examined in more varied cultural contexts. For instance, a multi-country study found that identity fusion to country in interdependent cultures like India, China, and Indonesia was higher than in independent countries like the US and Australia (Swann et al.,

2014). Nonetheless, even here the relationship with interdependence is not totally clear; fusion was surprisingly lowest in Japan – an interdependent culture – and fusion in Spain was the second lowest. There have also been cross-cultural studies using the pictorial measure. Purzychi & Lang (2019), which examined fusion in populations that were all presumably highly interdependent, found that rates ranged from 85% among the Hadza to 45% among Hindu Indo-Fijians. Together, our informal review of the literature suggests that interdependent cultures may be more prone to fusion, but that there is considerable variation.

In any case, although the pictorial measure allows participants to easily respond to multiple groups, we share your concern that responses can sometimes be hard to interpret. It is partially for this reason that we adopted the verbal measure in Study 2, which is also a better predictor of fusion outcomes (Varmann et al., 2024).

R1.6 Cooperation: *As willingness to cooperate was measured rather than actual cooperation, it would be better to have the title and narrative of the paper reflect this.*

Response: We have now changed the title of the paper to “Identity fusion can foster intergroup trust and *willingness to cooperate*” and clarified throughout the text that willingness to cooperate was measured (not actual cooperation).

Analyses:

R1.7 Model 5: *this is likely to be a simple reporting issue, but it says that “Fusion to regional group” is added as a main effect rather than “Fusion to political group” (or perhaps this is why Model 5 shows inconsistent results?)*

Response: Thank you for picking up on this mistake- this was a typo rather an analytic mistake. This has now been corrected to “political group” in Table 1.

R1.8 Aggregating across social groups: *As the regression analyses in Study 1 show, the status of the group (e.g., majority v minority) affects trust directly. Most theories of intergroup relations account for the finding that high- and low-status groups perceive the world and act within in it rather differently. However, the social environment that is highlighted in the introduction is not currently reflected in the analyses. Unfortunately, the reviewing files did not appear to include the supplementary materials mentioned in the document (and I could not open it on OSF either), so I have too little information on sample demographics (only gender for Study 1 and country for Study 2) to judge whether the sample consisted of equal numbers of high-status (e.g., Moro) and low-status group members (e.g., Christian settlers) - and it may therefore be feasible to analyse the data by groups - or whether results are dominated by majority group participants. Similarly, in Study 2, participants were aggregated across countries and irrespective of whether they are part of a religious majority or minority within their country. I wonder whether separate analyses would yield stronger, more interpretable findings.*

Response: We certainly agree that status can influence how intergroup relationships are perceived. In fact, we included a term for ingroup (e.g., Moro or Lumad) in each regression in Study 1 to control for any dispositional ingroup characteristics (such as status) that might influence the outcome measure.

It is difficult to ascertain which groups are high or low status in a complicated region like the BARMM. For example, the Moro could be considered higher status than Christian Settlers in the BARMM in that they are a majority of the local population, but also lower status in the broader context of a predominately Christian Philippines. Hence, the question of status can be difficult to judge a-priori.

We happen to have measured several items that can help us determine which groups are high or low status, particularly in terms of Governmental influence. For example, we asked participants questions related to the degree to which their ingroup's is helped by the BARMM Government ($\alpha = .75$):

1. *If [regional ingroup] need help, the BARMM Government will do its best to help them.*
2. *The BARMM Government acts in the interest of [regional ingroup].*
3. *The BARMM Government is genuinely interested in the wellbeing of [regional ingroup].*

(1 = Strongly disagree, 7 = Strongly agree)

We also asked questions about personal status. Participants were asked about the extent their voice is heard at the following levels of Government ($\alpha = .82$):

4. *To what extent do you feel your voice is heard by Government at the following levels:*
 - a. *Baranagy* (NB: removed from final measure to improve Cronbach's alpha)
 - b. *Municipality/City*
 - c. *BARMM*
 - d. *National Government*

(1 = Not at all heard, 7 = Very much heard)

And were asked to rank their subjective status:

5. *Imagine that this ladder pictures how society is set up. At the top of the ladder are the people who are best off – they have the most money, they highest amount of schooling, and the jobs that bring the most respect. At the bottom are the people who are worst off – they have the least money, little or no education, no job or jobs that no one wants or respects.*

Now think about yourself. Please tell us where you think you are on this ladder.

(1 = bottom of the ladder, 10 = top of the ladder).

We ran one-way ANOVAs to examine how the regional groups differed on these three measures. All ANOVAs were significant, although post-hoc contrasts were only significant for “Voice Heard” and “Subjective Status”. The group means (SD) are summarized below, with Tukey Pairwise Multiple Comparisons used to detect between group differences:

	Moro	Lumad	Christian Settler
BARMM Gov Help	6.08 (1.05)	5.86 (1.37)	5.88 (1.15)
Voice Heard	3.76 (1.22)*	4.17 (0.90)	4.06 (0.94)
Subjective Status	4.43 (1.48)	3.49 (1.49)*	4.46 (1.30)

Note. * Indicates a significant difference with both other groups

These results paint a mixed picture, with all groups inclined to believe that the BARMM Government helps their ingroup. Moro reported that their voices were heard the least, and Lumads reported the lowest subjective status.

As suggested, we also conducted the regression analyses with each group separately:

Ingroup	Outgroup	Fusion	Perceptions	Interaction
Christian	Muslim	0.62 (.15)**	0.09 (0.11)	0.12 (0.13)
Muslim	Christian	-0.13 (0.13)	0.13 (0.09)	0.53 (0.11)**
Lumad	Moro	0.06 (0.23)	-0.06 (0.12)	0.35 (0.15)*
Moro	Lumad	0.35 (0.18)	0.08 (0.15)	0.69 (0.17)**
Christian Settler	Moro	0.71 (0.14)**	-0.18 (0.09) *	0.39 (0.11)**
Moro	Christian Settler	0.37 (0.17)*	-0.03 (0.15)	0.77 (0.17)**
Christian Settler	Lumad	0.44 (0.15)**	-0.14 (0.11)	0.57 (0.15)**
Lumad	Christian Settler	-0.19 (0.27)	-0.04 (0.18)	0.69 (0.20)**
UBJP	BAPA	0.75 (0.12)**	0.32 (0.07) **	0.33 (0.12)**

Note. Estimate (std. error); * is significant at .05 level; ** is significant at the .01 level.

These analyses suggest that the interaction remains for all ingroups, excluding Christianity which only had a significant positive main effect for fusion. Given the results largely conform to our prior analyses we have not included them in our manuscript or supplementary materials, but would be open to doing so if the reviewer felt it was important.

Apologies for any difficulties accessing the OSF. We double-checked and the supplementary materials appear to be accessible via the OSF, so please let us know if you are still having technical difficulties. In any case, please find the sample split for each dyad in Table 1 below. We collected over 200 participants for all ingroups, excluding BAPA supporters (which were hence excluded from the above analyses).

Table 1

Intergroup dyad sample sizes

Social Identity	n	% of social identity category
Religious Identity		
Christian	269	34.27
Muslim	516	65.73
Other	31	3.80
Regional/Ethnic Group		
Moro	387	47.43
Lumad	207	25.37
Christian Settler	221	27.08
Other	1	0.12
Political Identity		
UBJP	471	57.7
BAPA	64	7.84
Other	281	65.56

Unfortunately, we cannot replicate these separate analyses for Study 2 as the ingroup for almost all participants was the most dominant (i.e., high status) religion in their country (Islam in Gambia and Pakistan, Christianity in Uganda and Tanzania).

R1.9 *Please also provide Study 1 correlations as you do for Study 2*

Response: We now provide the table below, which outlines the correlations between the key variables for each intergroup dyad (pg. 13):

Table 1

Zero-order correlations for key variables

Notes: CS is the abbreviation of “Christian Settler”. The reference group for the fusion term in all models was non-fused group members. Only correlations significant at the .05 level are displayed. Darker red implies a stronger negative correlation and darker blue implies a stronger positive correlation.

R1.10 Figures: *y*-axes were not consistent with the full scale and missing for Study 2

Response: We have now revised figures 2 and 3 so their *y*-axes cover the full scale. We also ensured a *y*-axis was present for the Study 2 figure (i.e., Figure 3), but please tell us if we misunderstood this part of your comment.

Thank you for the opportunity to review this fascinating paper and I hope my comments come across as a constructive contribution to strengthen the conclusions of the paper.

Miriam Koschate

Reviewer #2 (Remarks to the Author):

Review Comments

R2.1 *Limitations of Measurement Methods and Design*

A major concern is that both the dependent variables—“outgroup trust” and “outgroup cooperation”—and the moderator “perception” appear to capture aspects of outgroup positivity, which may lead to some conceptual redundancy. This raises the possibility that the interaction effects may partly reflect measurement overlap rather than distinct psychological processes. Although the analyses demonstrate that several variables act as moderators, it remains unclear which specific facet of outgroup positivity is the core driver of the moderating effect. Future research should aim to more clearly separate these constructs and provide robust empirical evidence and theoretical mechanisms to pinpoint the primary contributor.

Response: We thank the reviewer for this comment and agree that there is some overlap with the dependent cooperation/trust variables and the outgroup perception moderator. However, it is worth noting that the correlation between outgroup cooperation and perceptions in Study 2 was only moderate ($r = .49$), suggesting that the two measures captured different constructs. We have now discussed this in the limitations section (pg. 24):

“Finally, the dependent variables (i.e., trust and willingness to cooperate) may have some conceptual overlap with some moderators (i.e., outgroup perceptions), however the moderate correlation between them suggests that they are likely measuring distinct constructs.”

We also discuss possibilities for future research in line with your suggestions (pg. 24):

“Future research should also work to disentangle the relative contributions of the different facets of outgroup perceptions to determine the core driver of the moderating effect. This could involve experimentation in which each facet is directly manipulated.”

R2.2 *Theoretical Background and Hypothesis Development*

The manuscript is intriguing in its suggestion—based on the “fusion-secure base hypothesis”—that a strong sense of ingroup affiliation can also foster trust and cooperation toward outgroups. This is an important and timely contribution to the literature. However, the discussion would benefit from a more detailed comparison with previous research on identity fusion as well as traditional social identity theory. Such discussion would help clarify the novelty and distinctiveness of the hypotheses. Given the well-documented link between identity fusion and pro-group extreme behavior (i.e., outgroup aggression), a deeper exploration of the underlying psychological processes that account for both violent and cooperative behavior would strengthen the theoretical framework. In particular, while the mechanism centered on threat is discussed, the absence of a threat moderation effect in Study 2—in contrast to the moderation observed for perception—warrants further careful interpretation and discussion.

Response: We agree with your suggestion and have added considerably more theoretical context regarding the fusion-secure base hypothesis, particularly in terms of contrasting identity fusion theory with the social identity approach. We have now addressed this part of your request in response to a similar comment from Reviewer 1 (see response to R1.1).

Specifically, we now directly link the present results to past findings from identity fusion and social identity theory in the discussion, as well as more comprehensively explain how identity fusion could promote both intergroup cooperation and conflict.

We also agree that the absence of a threat moderation in both studies warrants careful consideration. We currently provide the following paragraphs in the general discussion (pg. 21-22), but please let us know if you feel there is an aspect that is currently lacking:

“It is worth considering why our scales explicitly measuring threat did not moderate the relationship as expected. One possibility is that the more overtly worded threat items were too blunt a tool to perform as intended. In both studies, but especially Study 1, the threat measures were very skewed towards perceiving the outgroup as non-threatening. While good news for the BARMM – perhaps reflecting decreased regional tensions 30 – this meant that these scales may have only captured extremely negative outgroup attitudes and missed more subtle gradations in outgroup perceptions. Moreover, explicit threat scales can only capture the negative side of outgroup perceptions (i.e., no threat vs. threat), whereas more general outgroup measures assess the full spectrum of attitudes (e.g., hostile vs. friendly) and so can better detect the relevant inflection point at which fusion begins to predict a willingness to cooperate. It is also worth noting that perceptions the outgroup had historically mistreated the ingroup (i.e., represented a past threat) did significantly moderate the relationship in Study 2, perhaps because it represents a more socially desirable threat measure.

More interestingly, these results may suggest that the original fusion-secure base hypothesis, which emphasized outgroup threat as the prime moderator, is overly simplistic. This possibility is reinforced by the clear moderating effect of the potential harms and benefits of cooperation itself in Study 2, highlighting that the nature of a specific relationship may matter as much or more than whether the outgroup is generally seen as threatening. History is littered with examples of sworn-enemies working together for the “greater good”, with the momentary benefits of cooperation seemingly outweighing the outgroup’s threat-status. Nevertheless, given that perceptions of the outgroup as a threat and viable cooperative partner are probably closely associated, this distinction may not always be important.”

R2.3 Interpretation of Results and Consistency in Discussion

The findings from both Study 1 and Study 2 generally support the proposed hypotheses, which is highly commendable given the complexity of the topic and the large-scale, international data utilized. However, since the measures for “perception” and “threat” regarding outgroups and the dependent variables “trust” and “cooperation” all assess aspects of outgroup positivity, there is potential overlap in what is being measured. This raises the possibility that the observed interaction effects might be influenced by shared measurement variance rather than distinct underlying processes. In particular, the fact that interaction effects emerged only for perception and not for threat suggests that the results might partly be driven by common measurement elements. This issue should be acknowledged and discussed to provide a more nuanced interpretation of the findings.

Response: Thank you- we acknowledge that this is a potential problem. We hope we have adequately addressed this issue in response to R2.1, but would be willing to extend our discussion if needed.

Overall Evaluation

Overall, this manuscript is an important contribution to the field, offering valuable insights into how identity fusion may promote positive intergroup relations using large-scale, cross-cultural data. The innovative approach of examining multiple interaction effects is particularly commendable. While there are areas for improvement—such as deepening the theoretical framework, refining measurement methods, and enhancing the rigor of statistical analyses—the study’s strengths and its potential to advance our understanding of complex intergroup dynamics are clear. Addressing the methodological challenges related to overlapping constructs will further enhance the impact and clarity of this promising work.

Thanks again to the reviewers for their assistance.

Sincerely,

Jack W. Klein

Brock Bastian

Emmanuel N. Odjidja

Samhita S. Ayaluri

Christopher M. Kavanagh

Alimudin M. Mala

Harvey Whitehouse